# *Cryptoblabes gnidiella* Millière (Pyralidae, Phycitinae): An Emerging Grapevine Pest in Greece

**DOI:** 10.3390/insects16010063

**Published:** 2025-01-10

**Authors:** Konstantinos B. Simoglou, Iraklis Topalidis, Dimitrios N. Avtzis, Achilleas Kaltsidis, Emmanouil Roditakis

**Affiliations:** 1Department of Quality and Phytosanitary Inspections, Rural Economy and Veterinary Directorate, 66133 Drama, Greece; simoglouk@pamth.gov.gr; 2Department of Agriculture, School of Agricultural Sciences, Hellenic Mediterranean University, Estavromenos, 71410 Heraklion, Greece; 3Industry Advisory Services, 66100 Drama, Greece; iratopalidis@gmail.com; 4Forest Research Institute, Hellenic Agricultural Organization Demeter, Vassilika, 57006 Thessaloniki, Greece; dimitrios.avtzis@fri.gr (D.N.A.); akaltsi@agro.auth.gr (A.K.); 5Institute of Agri-Food and Life Sciences, Hellenic Mediterranean University Research Centre, 71410 Heraklion, Greece

**Keywords:** honeydew moth, Mediterranean Basin, pest management strategies, Greece, wine-producing areas

## Abstract

A new pest has been found in Greek vineyards, specifically affecting a variety of grape called ‘Xinomavro’. The honeydew moth, a small insect native to the Mediterranean region, has been reported in other parts of Europe, the Middle East, North Africa, and South America, possibly due to global warming. In this study, we documented the first cases of honeydew moth infestation in two organic vineyards in northeastern Greece. Interestingly, the infestation only occurred in the ‘Xinomavro’ grape variety, which is harvested later in the year, while other grape varieties were not affected. This suggests that previous infestations in Greek vineyards may have been misidentified, and the actual impact of the honeydew moth may be underestimated. Our findings are important for grape growers and wine producers in Greece, as they highlight the need for better management and monitoring of this pest to prevent potential damage to crops and the wine industry.

## 1. Introduction

The subfamily Phycitinae (Pyraloidea, Pyralidae) comprises significant pests, with their larvae exhibiting considerable variation in their feeding habits. For instance, some of the most well-known species of this subfamily infest stored products, such as the tobacco moth *Ephestia elutella* (Hübn.), the flour moth *Anagasta kuehniella* (Zell.), the fig moth *Cadra cautella* (Wlk.), and the mealworm moth *Plodia interpunctella* (Hübn.) [1,2]. Other, equally notorious, pests include the locust bean moth *Apomyelois ceratoniae* (Zeller) feeding on carob and pomegranate [3], the pulse pod borer moth *Etiella zinckenella* (Treitschke), which is a pest of soybean [4], and the quince moth *Euzophera bigella* (Zeller), which infests olives [5]. In total, Powell [6] suggests that approximately two thirds of the 6000 described pyralid moth species (Pyralidae) are classified within the Phycitinae subfamily.

To date, the most important grapevine pests reported in Greece include two species of Lepidoptera, the European grapevine moth *Lobesia botrana* (D. and Schiff.) (Tortricidae, Olethreutinae) [7,8] and the grape berry moth *Eupoecilia ambiguella* (Hübn.) (Tortricidae, Tortricinae); several hemipteran species, including the grape mealybug *Planococcus ficus* (Signoret) (Pseudococcidae) [9,10]; the leafhoppers *Arboridia adanae* (Dlabola), *Asymmetrasca decedens* (Paoli), *Hebata decipiens* (Paoli), *Hebata vitis* (Göthe), *Jacobiasca lybica* (Bergevin), and *Zygina rhamni* Ferrari (Ciccadellidae, Typhlocybinae) [11]; and one thrips species, namely the western flower thrips *Frankliniella occidentalis* (Pergande) (Thripidae) [12,13].

Among members of the Phycitinae subfamily, the honeydew moth *Cryptoblabes gnidiella* Millière is the most prevalent and economically damaging species pest of grapevine in the coastal regions of Italy, France, and Israel [14,15,16,17]. In European vineyards, *C. gnidiella* was never considered a primary pest of the grapevine, likely due to its typically low population density and the fact that its presence in grape clusters, and thus the associated damage, was primarily attributed to other pests such as *L. botrana* [15,18].

*Cryptoblabes gnidiella* is native to the Mediterranean Basin, southern Europe, North Africa, and Southwest Asia. In Europe, the species is predominantly found in Portugal, Spain, Italy, Greece, Ukraine, and Austria [19,20,21]. The distribution of *C. gnidiella* in Europe is outlined by Karsholt and Razowski [22], who document its natural distribution in Norway, Denmark, Sweden, Finland, the Netherlands, the United Kingdom, France (including Corsica), Spain, Portugal, Italy (including Sardinia and Sicily), Malta, Austria, and Greece. On the contrary, records from the United Kingdom, the Netherlands, Scandinavian countries, and Poland are likely the result of interceptions and the accidental introduction of the species with imported fruits [15,19,21]. Finally, *C. gnidiella* has also been reported in Malaysia, New Zealand, Hawaii, several countries in Africa and Asia, as well as numerous tropical and subtropical regions of North and South America [15,21].

*Cryptoblabes gnidiella* develops on a variety of host plants, including citrus, grapes, plum, peach, apple, pear, medlar and loquat, fig, pomegranate, kiwi, mango, avocado, fejoia, blueberry, cotton, maize, carrot, persimmon, etc. [14,15,23,24,25]. Nevertheless, its larvae show a profound preference for grapes and citrus fruits [18]; in fact, larvae that feed on grapes develop faster than the ones that feed on citrus. Over its natural range, *C. gnidiella* is multivoltine, with six to seven generations in Israel [18,26], three to four generations in Italy and France [14], four generations in Antalya, Turkey [27], and three generations in Uruguay [28]. While the first generation is hardly causing any damage as the grapes are still unripe, the second generation causes the majority of damage since it coincides with the period of grape ripening [28]. Consequently, the impact of *C. gnidiella* is far more intense in late-ripening than in early-ripening cultivars [17]. To that, the notable rise in captures observed from the onset of grape ripening in Tuscany (Italy) is related to the outset of the second generation as well as to the movement of moths from adjacent vineyards driven by volatile compounds present in ripe and rotten grapes. The last two generations mostly overlap, leading to an increase in population density [14].

It has been demonstrated that the larvae of *C. gnidiella* are attracted to the juice of grape bunches previously affected by the grape berry moth, *L. botrana* [18]. To the same direction, Bagnoli and Lucchi [14] suggest that the irregular infestation in a vineyard is quite likely associated with factors that determine the availability of food resources, such as *L. botrana* infestation, *P. ficus* outbreaks (due to the presence of sugars and honeydew that young larvae consume), and disease-related deterioration of ripening grapes. However, it should also be noted that *C. gnidiella* larvae have the potential to cause damage to clusters that have not previously been affected by other pests, even prior to veraison [24]. This is attributed to superficial erosions on the rachis and peduncles of grapes that disrupt their vascular system [15]. *Cryptoblabes gnidiella* overwinters in the vineyard as non-diapausing larvae (1st–5th instar) concealed within the dried grape cluster remnants on the plant or on the ground [15,28]; yet, in late spring to early/mid-summer (April to July), adults migrate from the vineyard to alternative woody and herbaceous hosts for oviposition and offspring development [15]. This might well explain the inconsistency between the scarcity of the larval population from May to July against the high abundance of adults captured in July [15].

In the wine-producing area of the Regional Unit of Drama (Eastern Macedonia, Northeastern Greece), late infestation of grapes by lepidopteran larvae has frequently been attributed to *L. botrana*. In light of recent publications on the spread of *C. gnidiella* in Mediterranean viticultural areas, accompanied by reports of damage, a preliminary investigation has been conducted to identify the potential occurrence of *C. gnidiella* in wine grapevines in Drama. This brief communication presents the first observation of grape infestation by *C. gnidiella* in Greece.

## 2. Materials and Methods

### 2.1. Monitoring and Sampling Locations

One pheromone delta trap equipped with sticky plastic delta liners (Suterra^®^) utilizing Phero Norm^®^ lure was installed on the 20th of August 2024 in an organic vineyard (0.5 ha) located in the Kali Vrysi rural area (Drama, Greece) (41° 9.80′ N, 23° 53.03′ S) for the purpose of monitoring *C. gnidiella* on late-harvesting varieties, with the objective of investigating the presence or absence of the pest. Adults captures were recorded on a weekly basis. The mean, minimum, and maximum air temperatures were recorded on a daily basis from the meteorological station owned by the National Observatory of Athens (www.noa.gr, accessed on 20 August 2024), which is located in Mikrokampos (Drama, Greece) (41° 7.78′ N, 24° 0.73′ E) and situated 8 km away from the sampling point. Historical temperature data for the period between 2020 and 2023 have also been obtained from the same source. In the temperature data obtained, the average temperature for the period May-September 2024 was compared with the corresponding average temperature for the years 2020–2023 considered for the same period (May–September). For this purpose, the *t*-test was applied using the statistical programme Jasp 0.19.1 [29]. Additionally, specimens of infested grape bunches from the aforementioned plot and other adjacent vineyards were collected and transferred to the laboratory of the Department of Quality and Phytosanitary Inspections (Rural Economy and Veterinary Directorate, Drama, Greece) for further examination.

### 2.2. Species Identification

The process of species identification was conducted using both morphological traits and molecular tools. Morphological identification was based on adults collected from both sticky traps and infested grape bunches using species-specific taxonomic identification keys [15,23]. DNA barcoding was performed with larvae collected from infested bunches, which were immediately stored in 95% alcohol. In total, DNA was extracted from the body of three (3) larvae using PureLine^®^ Genomic DNA kit (Invitrogen, Waltham, MA, USA), following the manufacturer’s protocol. Polymerase chain reaction was run in 25 μL volume with LCO-HCO primers [30] that amplify a 658 bp long locus of the mitochondrial cytochrome oxidase subunit I (COI) gene. Concentrations and conditions of thermocycling are provided in Avtzis et al. [31]. PCR products were cleaned up enzymatically using the ExoSAP-IT^TM^ PCR Product Cleanup Reagent (ThermoFisher Scientific, Waltham, MA, USA) and then shipped to Cemia Company (Larissa, Greece), where they sequenced in an ABI3730XL automated sequencer using the same primers as in PCR. Sequences were initially visualized with Chromas Lite version 2.6.6 software and then blasted in the NCBI GenBank database.

### 2.3. Assessment of the Damage Extent

The level of damage was assessed immediately after the harvest, in early September, and particularly on the late ripening vines, especially the ‘Xinomavro’ cultivar, in early October. At least five hundred bunches per field were visually examined for symptoms of feeding damage, such as holes or hollow grapes, evidence of the presence of larvae, and the extent of deterioration of the grapes. Bunches deemed to be adversely affected by these factors were then segregated, and the estimated percentage of the total affected was calculated.

## 3. Results

### 3.1. Species Identification

Based on the comprehensive morphological keys of Neunzig [23] and Lucchi et al. [15], adults were readily identified as *Cryptoblabes gnidiella*. This finding was also validated by DNA barcoding, as the obtained sequences matched perfectly (100% resemblance) with MG895658 and by 99.85% with OQ564198, both of which correspond to *C. gnidiella*.

Adult exhibits a wing length of 5.0–6.5 mm [23]. The fore wings are dark grey, punctuated by tiny black spots, veiled in white, dotted with reddish scales, and characterised by indistinct lighter bands. The hind wings are shiny white and streaked with terminal grey lines [24]. The proboscis and labial palps of the adults are notably elongated, a trait shared by the majority of Phycitinae. The antennae are characterised by a simple structure and a fine covering of cilia. In males, a distinctive horn-shaped projection is present on the third antennal segment, which serves as a key taxonomic character [15] (Figure 1). Larvae exhibit a prominent, dark brown to black, pinaculum ring in conjunction with the SD1 setae on the mesothorax, along with a less pronounced pinaculum ring surrounding the SD1 setae on the eighth adnominal segment [23]. The fifth instar larva is 10–12 mm. The dorsal side of the body is characterised by a yellow to light brown colouration, with two narrow, longitudinal, darker bands [24] (Figure 2).

### 3.2. Monitoring of Flight in Pheromone Traps

The pheromones we used were highly species-specific, with adult captures peaking on 1 September (107 specimens/trap/week) and remaining high (>50 specimens/trap/week) throughout September (Figure 3).

The mean daily temperature recorded locally in the aforementioned sampling region from May to September 2024 (23.9 °C ± 0.3) was statistically significantly higher (*t*-test, *p* < 0.001) compared to the same months in 2020–2023 (22.6 °C ± 0.1). A similar trend was evinced in the mean minimum (16.1 °C ± 0.3 vs. 15.5 °C ± 0.2; *p* = 0.001) and maximum (32.0 °C ± 0.4 vs. 30.4 °C ± 0.3; *p* < 0.001) temperatures for the same period. Accordingly, the phenological stages of the ‘Xinomavro’ cultivar began with the onset of bud development on 20 March (BBCH 05), the end of flowering on 17 May (BBCH 69), and fruit set on 24 May (BBCH 71). The grape touching stage (BBCH 77) was observed on 16 June, followed by the beginning of ripening (BBCH 81) on 24 July. The harvesting process in ‘Xinomavro’ was initiated on 1 October.

### 3.3. Description of the Damage

The presence of frass and excrement deposits from the larvae was observed on infested grape bunches. The larvae’s feeding activity resulted in damage to the grapes, with visible signs of infestation, including hollow grapes, desiccation, and shriveling. The infested bunches were deteriorated due to the development of grey mold, rendering them unsuitable for vinification. Larval feeding damage compromised the overall quality of grapes, negatively impacting the desired key parameters for winemaking. The grape bunches were, therefore, deemed unsuitable and were discarded.

### 3.4. Estimation of the Damage Level

During the harvesting process, it was determined that the late-harvested grape cultivar ‘Xinomavro’ was the sole cultivar affected by the infestation in two vineyards located at Kali Vrysi (41° 9.80′ N, 23° 53.00′ E and 41° 10.10′ N, 23° 53.80′ E) (Figure 4). The level of infestation was estimated to be between 5% and 10% of the total number of bunches harvested from the vineyards of this particular cultivar. Significant labour costs were expended for the sorting of grapes during the harvesting process and immediately prior to initiating the vinification process. Prior to the beginning of the harvest, an additional sorting operation was conducted to remove infested bunches. It has been estimated that the harvesting rate was approximately 40% of the usual quantity harvested per hour per worker, due to the on-site sorting in the vineyard and the removal of the infested bunches.

## 4. Discussion

This is the first report of *C. gnidiella* feeding on grapevines in Greece. Specifically, infestations were observed in two organic vineyards, situated in a wine-producing area within the Regional Unit of Drama in Northeastern Greece. This finding triggers further research that is required to assess the potential impact of *C. gnidiella* on grapevine productivity in Greece, particularly in terms of yield, quality, and economic losses. Lucchi et al. [15] suggest that *C. gnidiella* has not been regarded a significant pest of grapevines in European vineyards, likely due to its generally low population density, coupled with the fact that damage in grape clusters was commonly (and likely erroneously) attributed to other more abundant pests, mainly *L. botrana*. As a consequence, it seems probable that the actual impact of *C. gnidiella* in Greece is underrated, as previous late summer infestations of lepidopteran larvae in vineyards were collectively considered to be *L. botrana* infestations.

The temperature conditions of the 2024 growing season have been observed to exert a distinctive influence on the phenological stages of the crop. Given that early ripening of grapevines in response to warm conditions is more likely to be caused by shifts in the onset of ripening [32], it is worth noting that the onset of ripening (BBCH 81) was reached in ‘Xinomavro’ on 24 July 2024, whereas it was reached on 20 August the previous year. The prevailing meteorological conditions resulted in an early completion of the harvest, with most of the region’s cultivars, including ‘Assyrtiko’, ‘Malagousia’, ‘Cabernet sauvignon’, ‘Sauvignon blanc’, ‘Limniona’, ‘Chardonnay’, and ‘Agiorgitiko’, being harvested at the end of August and beginning of September, several days earlier than the typical harvest period. Harvest in ‘Xinomavro’ initiated on 1 October, approximately one week before the anticipated date.

The impact of temperature on the progression and spread of infestations appears to be considerable. The mean, minimum, and maximum daily temperatures from May to September 2024 were, respectively, 1.3 °C, 0.6 °C, and 1.6 °C higher than those observed over the preceding four years (i.e., 2020 to 2023). Zumbado-Ulate et al. [33] demonstrate that the predicted climatic adaptability for *C. gnidiella* increases with elevated temperatures along the west coast of the United States.

The late-ripening grape cultivar ‘Xinomavro’ was the only cultivar affected, which is in consistency with previous studies. For example, clusters of the late-ripening ‘Gewürztraminer’ cultivar in Uruguay are severely affected at harvest time, whereas the ‘Pinot noir’ cultivar, which is harvested earlier, has demonstrated resilience to economic damage, even during years of high *C. gnidiella* population density [28].

A range of organic viticultural practices and biological control methods are used to manage infestations in vineyards. The latter includes mating disruption on a large scale mainly for the management of *L. botrana* and *P. ficus* populations [8,34,35]. Biological insecticides and other chemical substances permitted in organic farming are employed only in the event of a population outbreak. In 2024, neither *L. botrana* nor *P. ficus* were observed in the vineyards of Kali Vrysi, and mating disruption method was implemented exclusively for the former pest. In several Mediterranean coastal areas, the *C. gnidiella* has recently demonstrated a markedly elevated level of harmful activity, characterised by an increased frequency and intensity of infestations. While the underlying causes of this surge remain underexplored, Lucchi et al. [15] postulate that global warming trends and the implementation of mating disruption for the management of *L. botrana*, with a consequent reduction in the use of pesticides, may be contributing factors.

As the majority of carpophagous insects that infest grapevines, the feeding of *C. gnidiella* larvae increases the probability of affected bunches becoming infected by pathogenic rotting agents and other saprophagous insects (Drosophilidae and Nitidulidae). However, the actual extent of this phenomenon varies with microclimate and the phenological stage of grapes [15,24,25]. At present, within the Mediterranean zone, only *L. botrana* is associated with ochratoxin A (OTA) accumulation in grapes [36], without any concrete evidence that involves *C. gnidiella* in OTA contamination. Nevertheless, Mondani et al. [36] suggested that, given the analogous characteristics of their feeding behavior, it is likely that they could also exert a comparable influence on the enhancement of OTA contamination.

Several studies have been conducted to model and describe the phenological stages and to predict the damage caused by the pest. Ringenberg et al. [37] determined the development thresholds and thermal constants for larvae fed on artificial diet. Their findings indicated that *C. gnidiella* requires a minimum temperature of 12.26 °C for its development and a total of 570 degree-days (°C) to complete one generation (egg to adult). Using these parameters, Öztürk [38] assessed the thermal requirements for egg hatching of this species in pomegranates in Mersin (Turkey), and concluded that the first, second, third, fourth, and fifth generations require 250, 800, 1375, 1930, and 2500 degree-days (°C), respectively. It has also been established by Wysoki et al. [39] that the preovipositional period lasts for 24 h following mating. Subsequently, over 50% of the eggs are laid in the first four nights of the female’s lifespan, and the mean fecundity was 105 eggs per female. Finally, using the above-mentioned thermal parameters, Vidart et al. [28] studied the correlation between male catches and damage in Uruguay. Even though pheromone traps detect moths and predict larval feeding on clusters, they proved that damage is more closely related to the cultivar than to adult population monitoring. It is, therefore, essential that monitoring efforts are intensified in order to accurately assess the extent of damage caused by *C. gnidiella*. This should be based on both pheromone traps and visual inspections [15].

As previously proposed by Bagnoli and Lucchi [14], the effective protection of grapes from *C. gnidiella* necessitates the implementation of effective control measures for both *L. botrana* and *P. ficus*. When executed effectively, these measures can drastically reduce the need for pesticide applications targeting *C. gnidiella*. Furthermore, the application of *Bacillus thuringiensis* formulations can be additionally beneficial, particularly in organic viticulture, when there is an asynchronous outbreak of different pests [14]. However, the results have not always been consistent due to various reasons (e.g., insecticidal activity by ingestion, low rain resistance, and short residual activity) [15]. In the same direction, mating disruption stands as a promising strategy that needs to be further explored [16,26,40].

The impact of climate change on global viticulture is a complex and multifaceted challenge that requires a comprehensive approach to crop protection strategies. The impact of climate change on grapevine pest insects and their antagonists is becoming increasingly evident. Growers must adapt their plant protection practices to mitigate the risks posed by climate warming. One critical consideration is the effect of rising temperatures and altered precipitation patterns on grapevines, pests, and their natural enemies. The potential for certain pests to expand their geographic ranges or become more abundant in warmer conditions [41] and the possibility of asynchrony between the larvae-resistant growth stages of grapevine and pest [42] represent new threats to growers. Adaptations must, therefore, take these changing dynamics into account. Grower adaptations may include enhanced monitoring of pest populations and adjusting the timing of pesticide applications accordingly. Integrated pest management practices, which combine various control methods, can help growers reduce their reliance on chemical pesticides and develop more sustainable approaches. Additionally, collaboration between researchers, growers, and policymakers is crucial to develop effective and sustainable solutions for crop protection in the face of climate change.

## 5. Conclusions

The emergence of *C. gnidiella* as a pest in Greek vineyards highlights the need for increased awareness and research regarding its impact on grapevine productivity. The observed shifts in phenological stages due to rising temperatures underscore the complex interplay between climate change and pest dynamics. Effective management strategies, including integrated pest management and biological control methods, are essential to mitigate the risks posed by *C. gnidiella* and other pests. As climate change continues to alter the viticultural landscape, growers must adapt their practices to ensure sustainable grape production, emphasizing the importance of collaboration among researchers, growers, and policymakers to develop resilient crop protection strategies.

## Figures and Tables

**Figure 1 insects-16-00063-f001:**
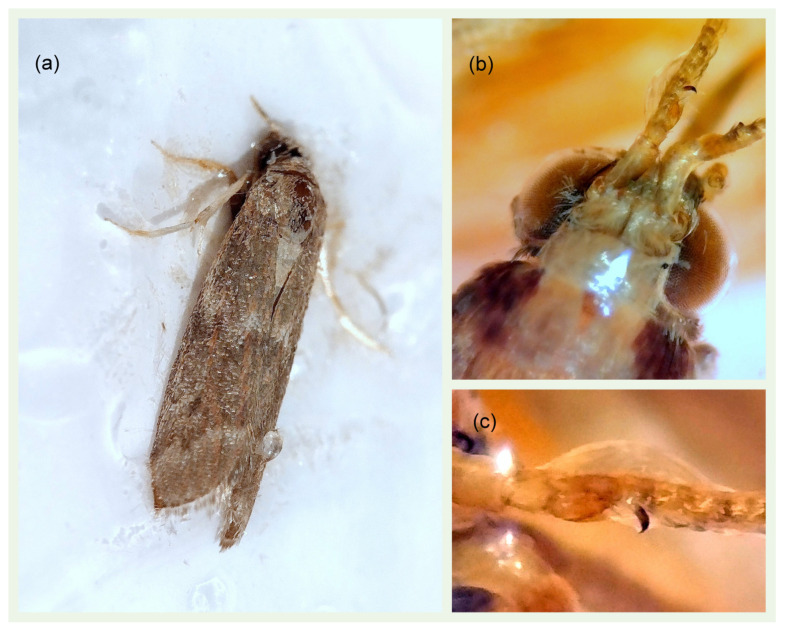
(**a**) Male individual of *Cryptoblabes gnidiella* caught on a pheromone sticky trap; (**b**) a horn-shaped projection is present on the males’ third antennal segment; (**c**) detail of the horn-shaped projection on the males’ third antennal segment.

**Figure 2 insects-16-00063-f002:**
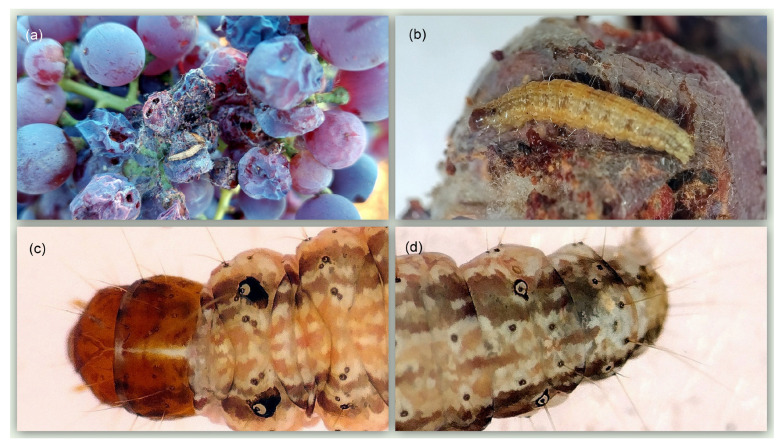
(**a**) Grape ‘Xinomavro’ bunch infested by *Cryptoblabes gnidiella* larvae; (**b**) *Cryptoblabes gnidiella* larva on grape; (**c**) dark pinacula in conjunction with the SD1 setae on the larval mesothorax; (**d**) dark pinacula surrounding the SD1 setae on the eighth abdominal segment of the larva.

**Figure 3 insects-16-00063-f003:**
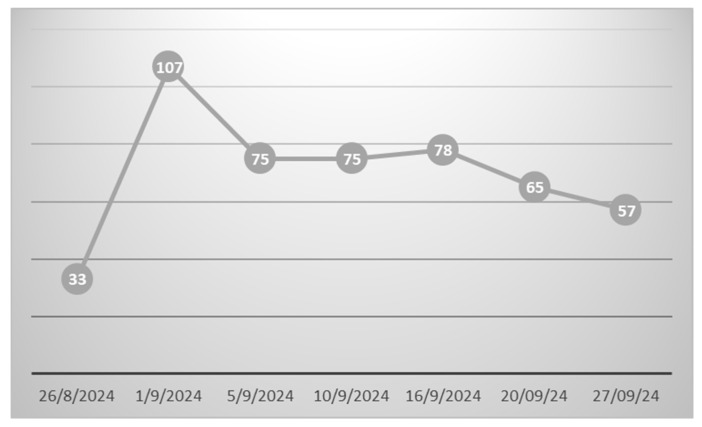
Schematic representation of the number of male *Cryptoblabes gnidiella* captures within a pheromone trap.

**Figure 4 insects-16-00063-f004:**
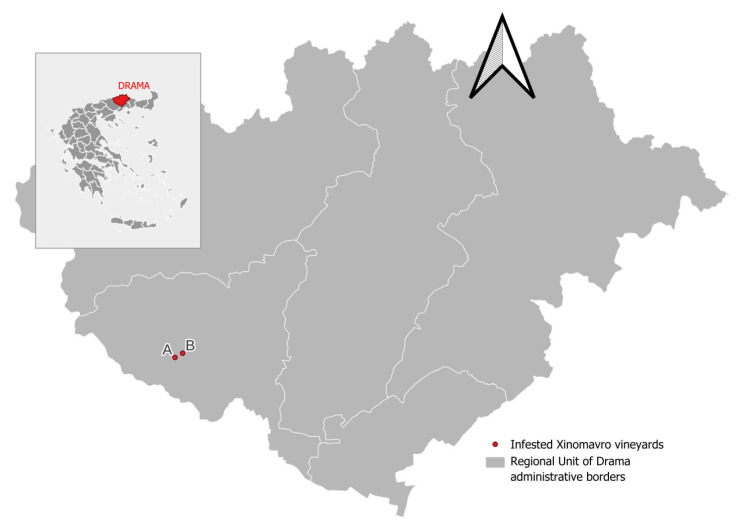
Locations of vineyards infested by *Crypboblabes gnidiella*; Kali Vrysi, Drama, Greece. A, B: Infested Xinomavro vineyards.

## Data Availability

The original contributions presented in this study are included in the article. Further inquiries can be directed to the corresponding author.

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
