# Peer review of "Cryptoblabes gnidiella Millière (Pyralidae, Phycitinae): An Emerging Grapevine Pest in Greece"

_insects, 2025, doi:10.3390/insects16010063_

Round 1
Reviewer 1 Report
Comments and Suggestions for Authors
Abstract:
The abstract needs to be more descriptive of some of the study's findings. For example, the late-ripening grape cultivar ‘Xinomavro’ was the only cultivar affected.
Introduction:
Ln 31, 32 & 40 - 45: Common names missing for the pests in their first mention.
Throughout the manuscript, use only scientific or common names for the pests. Keep them consistent.
Materials and Methods:
Ln 104: How big is the vineyard?
Is one pheromone trap enough to gauge pest population density across the vineyard?
Results:
Ln 154: Incorrect in-text citation. Adults are illustrated in Figure 2.
Ln 159: Incorrect in-text citation. Larvae instars are illustrated in Figure 1.
Ln 173: A table/line graph on pest densities over time would add value to the paper and make it easier for the reader to understand.
Author Response
Dear Reviewer 1,
Please see the attached file with authors' responses.
Yours sincerely,
Prof. Emmanuil Roditakis, Konstantinos B. Simoglou

Reviewer 2 Report
Comments and Suggestions for Authors
Dear Authors,
In the attached file, you will find some recommendations.
Regards,

Author Response
Dear Reviewer 2,
Please see the attached file with responses to your suggestions.
Yours sincerely,
Prof. Emmanuil Roditakis, Konstantinos B. Simoglou

Round 2
Reviewer 2 Report
Comments and Suggestions for Authors
Dear Authors,
Thanks for the report and for accepting my recommendations.
Regards,